# Detection Rates of PSMA-PET Radiopharmaceuticals in Recurrent Prostate Cancer: A Systematic Review

**DOI:** 10.3390/diagnostics15101224

**Published:** 2025-05-13

**Authors:** Soroush Rais-Bahrami, Phillip Davis, Albert Chau, Samuel J. Galgano, Brian F. Chapin, David M. Schuster, Catriona M. Turnbull

**Affiliations:** 1Department of Urology, University of Alabama at Birmingham, Birmingham, AL 35294, USA; 2Department of Radiology, University of Alabama at Birmingham, Birmingham, AL 35294, USA; samuelgalgano@uabmc.edu; 3O’Neal Comprehensive Cancer Center, University of Alabama at Birmingham, Birmingham, AL 35294, USA; 4Blue Earth Diagnostics Inc., Needham, MA 02494, USA; phillip.davis@blueearthdx.com; 5Blue Earth Diagnostics Ltd., Oxford OX4 4GA, UK; albert.chau@blueearthdx.com (A.C.); catriona.turnbull@blueearthdx.com (C.M.T.); 6Department of Urology, The University of Texas MD Anderson Cancer Center, Houston, TX 77030, USA; bfchapin@mdanderson.org; 7Division of Nuclear Medicine and Molecular Imaging, Department of Radiology and Imaging Sciences, Emory University, Atlanta, GA 30322, USA; dschust@emory.edu

**Keywords:** biochemical recurrence, prostate cancer imaging, prostate specific membrane antigen, positron emission tomography

## Abstract

**Background/Objectives**: To conduct a systematic review to evaluate the detection rates (DR) of the three FDA-approved PSMA-targeted radiopharmaceuticals in patients with recurrent prostate cancer. **Methods**: Two individuals systematically searched MEDLINE, ScienceDirect, and Cochrane Libraries (February 2025), and independently reviewed all results to identify studies reporting patient-level ^68^Ga-PSMA-11, ^18^F-DCFPyL, or ^18^F-flotufolastat DR in ≥100 evaluable patients with recurrent prostate cancer. Sample-weighted means (SWM) of extracted DR were calculated. **Results**: Of 5059 published articles, 37 met our inclusion criteria, reporting data from 8843 patients undergoing ^68^Ga-PSMA-11 (*n* = 27), ^18^F-DCFPyL (*n* = 8), or ^18^F-flotufolastat (*n* = 2) studies. Heterogeneity was noted across enrolled populations, particularly in prior treatments. ^68^Ga-PSMA-11 studies recruited patients with marginally higher median PSA than ^18^F-DCFPyL or ^18^F-flotufolastat studies (median PSA ranged from 0.1 to 10.7, 0.2–2.5, and 0.6–1.1, respectively). Reported overall DR ranged from 25 to 91% for ^68^Ga-PSMA-11, 49–86% for ^18^F-DCFPyL, and 73–83% for ^18^F-flotufolastat, with SWM of 71%, 66%, and 79%, respectively. Post-prostatectomy DR were reported in 18 articles, resulting in SWM DR of 58% for ^68^Ga-PSMA-11 (*n* = 12), 55% for ^18^F-DCFPyL (*n* = 4), and 76% for ^18^F-flotufolastat (*n* = 2). Among post-radiotherapy patients, SWM were 87% for ^68^Ga-PSMA-11 (*n* = 4), 90% for ^18^F-DCFPyL (*n* = 2), and 99% for ^18^F-flotufolastat (*n* = 1). SWM DR at PSA < 1 ng/mL were 53%, 42%, and 66% for ^68^Ga-PSMA-11 (*n* = 13), ^18^F-DCFPyL (*n* = 5), and ^18^F-flotufolastat (*n* = 2), respectively. **Conclusions**: Considerable heterogeneity exists across populations in studies of diagnostic PET radiopharmaceuticals. Despite a paucity of ^18^F-DCFPyL and ^18^F-flotufolastat studies compared with ^68^Ga-PSMA-11, the available data suggest that all three radiopharmaceuticals provide high overall DR in patients with biochemical recurrence of prostate cancer.

## 1. Introduction

Prostate cancer is the second most common cancer among men worldwide, with an estimated 299,010 new cases and 35,250 deaths in the United States in 2024 [1]. Despite recent advances in diagnosis and primary treatment, up to 53% of patients still experience biochemical recurrence, or rising serum prostate specific antigen (PSA) levels after primary, curative-intent therapy [2].

Early and accurate delineation of the location and extent of recurrent prostate cancer lesions with a sensitive imaging modality can help inform salvage therapy plans and improve patient outcomes [3]. In recent years, positron emission tomography (PET) with radiopharmaceuticals that target prostate-specific membrane antigen (PSMA) has become the mainstay for imaging patients with prostate cancer owing to its improved sensitivity and diagnostic performance compared with conventional imaging methods such as computed tomography (CT), magnetic resonance imaging (MRI), and bone scintigraphy [2,4].

Three radiopharmaceuticals (^68^Ga-PSMA-11, ^18^F-DCFPyL [^18^F-piflufolastat], and ^18^F-flotufolastat [^18^F-rhPSMA-7.3]) that target the PSMA receptor are US Food and Drug Administration (FDA)-approved for imaging patients with recurrent prostate cancer [5,6,7]. The two ^18^F-labeled radiopharmaceuticals (^18^F-DCFPyL and ^18^F-flotufolastat) are manufactured with use of a cyclotron and have a half-life of approximately 110 min, while the ^68^Ga-labeled agent, which requires the use of a generator, has a shorter half-life of 68 min [5,6,7]. All show similar biodistribution patterns, with the lacrimal and salivary glands, small bowel, liver, spleen, kidney and bladder being the normal organs typically showing the highest absorbed doses [5,6,7,8,9]. All three are renally excreted, but there is some evidence to indicate lower urinary activity of ^18^F-flotufolastat at the time of imaging compared with ^68^Ga-PSMA-11 and ^18^F-DCFPyL [10] which may result in improved image assessment. However, there are no direct head-to-head comparisons of the diagnostic performance of these three PSMA-targeted radiopharmaceuticals in the same patients with recurrent prostate cancer. Moreover, new FDA requirements have seen the introduction of novel endpoints for the assessment of new diagnostic imaging agents that were not utilized in trials of earlier approved agents [11]. As such, cross-study comparisons of primary endpoints can be challenging [12]. However, the overall patient-level detection rate, without verification of imaging findings, still represents one of the main endpoints traditionally reported in diagnostic imaging studies of prostate cancer. Therefore, we conducted a descriptive systematic literature review to evaluate the overall patient-level detection rates of the three FDA-approved PSMA-targeted radiopharmaceuticals in order to evaluate their performance in patients experiencing biochemical recurrence of prostate cancer following curative-intent treatment.

## 2. Materials and Methods

This systematic review adhered to the Preferred Reporting Items for Systematic Reviews (PRISMA) guidelines [13]. Meta-analysis was not possible therefore Synthesis Without Meta-analysis (SWiM) guidance [14] was followed. Scientific literature databases (MEDLINE, ScienceDirect, and Cochrane Libraries) were systematically searched in October 2023 and again in February 2025 as part of a wider review using the search terms detailed in Figure 1. Searches were limited to journal articles published in English in the preceding 10 years. All review articles, practice guidelines, case reports, editorials, and letters were excluded. Studies that reported data from fewer than 100 patients in the cohort of interest were excluded to avoid the variability in data from small sample sizes. We also excluded any studies that included patients with PSA persistence following primary therapy among their cohort of patients with biochemical recurrence. A minimum data requirement was for a patient-level ^68^Ga-PSMA-11, ^18^F-DCFPyL, or ^18^F-flotufolastat detection rate in 100 or more patients with biochemical recurrence of prostate cancer. All included studies had to either report this directly in the article, or to have included data from which it could be extrapolated.

Following removal of any duplicate articles, the search results were screened first based on the title and then on the abstract text outlined in Figure 1. The full text of all remaining articles was then reviewed to identify all reports meeting our pre-set criteria. Where multiple reports from the same study population were identified, the report with the largest population of interest was selected for review. Retrospective analyses that included whole or partial datasets already selected for inclusion were omitted. The article screening was performed by two independent evaluators, and any discrepancies were resolved by discussion with the wider author group. Manual searches of the reference lists of all included articles were conducted using the same screening process as above to seek further studies for inclusion.

The full text of all included studies was reviewed, and any patient-level ^68^Ga-PSMA-11, ^18^F-DCFPyL, or ^18^F-flotufolastat detection rates were extracted along with key data regarding patient baseline characteristics and clinical history for the cohort of interest. Any available data that allowed determination of patient-level detection rates among patients with low PSA values (<1 ng/mL) or stratified by initial therapy (post-prostatectomy or post-radiotherapy) were also extracted, along with any positive predictive value (PPV) estimates. Sample-weighted means were calculated to appropriately weigh the detection rate reported in each individual study by its sample (cohort) size.

Studies were assessed for risk of bias using the Quality Assessment of Diagnostic Accuracy Studies-2 (QUADAS-2) tool which assesses studies based on four domains: patient selection, index test, reference standard, and the study flow and timing [15].

## 3. Results

### 3.1. Search Results

The literature searches identified 5059 unique articles. Of these articles, 37 articles met our inclusion criteria and were considered suitable for evaluation in this systematic review (Figure 1). Thirteen (35%) of the articles reported prospective studies, 23 (62%) reported retrospective studies, and one (2.7%) reported an ambispective study. In total, 27 (73%) articles reported data with ^68^Ga-PSMA-11 [16,17,18,19,20,21,22,23,24,25,26,27,28,29,30,31,32,33,34,35,36,37,38,39,40,41,42], eight (22%) with ^18^F-DCFPyL [43,44,45,46,47,48,49,50], and two (5.4%) with ^18^F-flotufolastat [11,51].

Assessment of bias using the QUADAS-2 tool is summarized in Table 1. Patient selection and the index test methodology were broadly considered to be at low risk of bias, although some studies introduced limits on PSA levels or a requirement for negative conventional imaging which may have introduced a source of selection bias. When considering the reference standard domain, all studies were considered to have high risk. Histopathology, the gold standard for verification of positive PET scans was not routinely utilized.

### 3.2. Patient Populations

Table 2 summarizes the patient populations of the 37 studies reported by the articles included in our review. In total, data from 8843 patients with biochemical recurrence who were scanned with ^68^Ga-PSMA-11 (*n* = 6497), ^18^F-DCFPyL (*n* = 1715), or ^18^F-flotufolastat (*n* = 631) were evaluated across the 34 studies.

The inclusion criteria for defining biochemical recurrence were not well reported by many of the included articles. Twelve (32%) articles included no details on how patients were defined as having biochemical recurrence. Eleven (30%) indicated that inclusion was based on some level of PSA rise post-therapy, while the remaining 14 (38%) studies selected patients who met the RTOG-ASTRO Phoenix consensus criteria definition of biochemical recurrence [52].

Nineteen of the 34 studies (51%) reported the mean age of their cohort, and these ranged between 61 and 71 years. A median age of the cohort was reported for 15 of the studies (41%), ranging from 66 to 75 years, with the remainder of reports not including age data.

Median PSA levels were reported in 32/37 (86%) studies. These ranged from 0.1 to 10.7 ng/mL. ^68^Ga-PSMA-11 studies recruited patients with marginally higher median PSA levels than ^18^F-DCFPyL or ^18^F-flotufolastat studies; the reported median PSA ranged from 0.1 to 10.7 ng/mL for ^68^Ga-PSMA-11, 0.2–2.5 ng/mL for ^18^F-DCFPyL, and 0.6–1.1 ng/mL for ^18^F-flotufolastat.

Most of the evaluated cohorts comprised a mixed population of patients initially treated with radical prostatectomy and/or radiotherapy. Thirteen (35%) articles (ten ^68^Ga-PSMA-11, two ^18^F-DCFPyL and one ^18^F-flotufolastat) studied cohorts of post-prostatectomy patients only and two (5.4%; both ^68^Ga-PSMA-11) studied cohorts of only patients who had not undergone prostatectomy (Table 2). Taken together, across all the articles, approximately 78% of the evaluable patients had undergone radical prostatectomy prior to PSMA-PET. When stratifying this by radiopharmaceutical, approximately 76% of patients undergoing ^68^Ga-PSMA-11 PET, 79% undergoing ^18^F-DCFPyL PET, and 87% undergoing ^18^F-flotufolastat PET were post-prostatectomy.

### 3.3. Detection Rates

#### 3.3.1. Overall Patient-Level Detection Rates

Across the 27 ^68^Ga-PSMA-11 studies, the overall patient-level detection rate ranged from 25 to 91%. The overall patient-level detection rate ranged from 49 to 86% across the eight ^18^F-DCFPyL studies and from 73 to 83% for the two ^18^F-flotufolastat studies. The sample-weighted mean detection rates calculated across all studies are 71% for ^68^Ga-PSMA-11, 66% for ^18^F-DCFPyL, and 79% for ^18^F-flotufolastat (Figure 2). Breakdown of the detection rates by anatomical region was not consistently reported for all radiopharmaceuticals.

#### 3.3.2. Patient-Level Detection Rates Stratified by Prior Treatment

Eighteen articles (49%; 12 ^68^Ga-PSMA-11 [18,19,21,24,26,27,30,35,37,38,39,41], four ^18^F-DCFPyL [44,46,48,50] and two ^18^F-flotufolastat [11,51]) reported detection rate data for patients who had undergone radical prostatectomy. The overall patient-level detection rate among patients who had undergone radical prostatectomy ranged from 25 to 80% across the 12 ^68^Ga-PSMA-11 studies, from 47 to 76% across the four ^18^F-DCFPyL studies, and from 73 to 78% across the two ^18^F-flotufolastat studies. The sample-weighted mean detection rates patients who had undergone radical prostatectomy are 58% for ^68^Ga-PSMA-11, 55% for ^18^F-DCFPyL, and 76% for ^18^F-flotufolastat as shown in Figure 3.

Fewer articles (*n* = 7 [19%]; four ^68^Ga-PSMA-11 [24,25,37,40], two ^18^F-DCFPyL [46,48], and one ^18^F-flotufolastat [11]) reported detection rate data for patients who had undergone primary radiotherapy. The overall patient-level detection rate among those who had undergone radiotherapy ranged from 67 to 97% across the four ^68^Ga-PSMA-11 studies, 89–92% across the two ^18^F-DCFPyL studies and was reported to be 99% in the only ^18^F-flotufolastat study. Sample-weighted mean detection rates for patients who had undergone radiotherapy calculated from these data are 87% for ^68^Ga-PSMA-11, 90% for ^18^F-DCFPyL, and 99% for ^18^F-flotufolastat as presented in Figure 3.

#### 3.3.3. Patient-Level Detection Rates Stratified by PSA Levels

In total, 20 of the 37 (54%) included articles reported detection rate data for patients who had a prescan PSA level < 1 ng/mL. Of these, 13 reported ^68^Ga-PSMA-11 data [16,23,24,26,28,29,31,33,34,38,40,41,42], five ^18^F-DCFPyL data [43,44,46,47,48], and two ^18^F-flotufolastat data [11,51]. The overall patient-level detection rate among patients with a PSA level < 1 ng/mL ranged from 30 to 73% across the 13 ^68^Ga-PSMA-11 studies, from 38 to 58% across the five ^18^F-DCFPyL studies, and from 64 to 68% across the two ^18^F-flotufolastat studies. The sample-weighted mean detection rates in patients with a PSA level < 1 ng/mL are 53% for ^68^Ga-PSMA-11, 42% for ^18^F-DCFPyL and 66% for ^18^F-flotufolastat as shown in Figure 4.

Figure 5 provides a summary of the sample-weighted mean detection rates of ^68^Ga-PSMA-11, ^18^F-DCFPyL and ^18^F-flotufolastat among patients with biochemical recurrence of prostate cancer to illustrate the impact of low PSA level and prior treatment.

### 3.4. Positive Predictive Values

Only four (11%) of the studies verified the PSMA-positive detections as true and false positive in order to estimate the patient-level PPV [11,17,28,47]. Three of these studies provided PPV estimates based on histopathological verification of PSMA-positive lesions and show the PPV to be 84% for ^68^Ga-PSMA-11 [28], 79–83% for ^18^F-DCFPyL [47], and 82% for ^18^F-flotufolastat [11]. The fourth study used a composite reference standard that comprised PSA- or radiological-response to treatment, confirmative histology, or correlative imaging to verify ^68^Ga-PSMA-11-positive lesions and reported a higher PPV estimate of 93% [17].

## 4. Discussion

The advent of PSMA-targeting PET radiopharmaceuticals has facilitated the localization of small lesions below the resolution of conventional imaging techniques, earlier in the recurrence timeline. To date, three PSMA-targeting PET radiopharmaceuticals have been FDA-approved for use in recurrent prostate cancer. We conducted a large-scale systematic literature review to evaluate evidence of their detection efficacy in patients with recurrent prostate cancer.

Our data show the overall patient-level detection rate to be high for all three PSMA-targeted radiopharmaceuticals, with sample-weighted mean detection rates ranging from 71% for ^68^Ga-PSMA-11, to 66% for ^18^F-DCFPyL, and 79% for ^18^F-flotufolastat. However, considerable heterogeneity was noted across the enrolled populations, which likely would have influenced the detection rates. For example, a patient’s PSA level is known to impact the detection rate of all three radiopharmaceuticals, with higher detection rates observed at higher PSA levels [23,47,53]. Our analysis highlights the variation in the median PSA levels of the enrolled cohorts, with a broader range of median PSA levels noted for ^68^Ga-PSMA-11 studies than for studies of the other two radiopharmaceuticals.

Currently, biochemical recurrence is defined using the RTOG-ASTRO Phoenix consensus criteria [52]. However, as noted previously by others, these criteria have limited sensitivity for identifying early relapse that may require treatment before the Phoenix PSA thresholds are met [25]. It is likely that these thresholds, which consider the limited sensitivity of conventional imaging techniques such as CT, MRI, and bone scintigraphy, need redefining in the PSMA era. As such, it is perhaps not surprising that only 38% of the evaluated studies selected patients who met the RTOG-ASTRO Phoenix consensus criteria definition of biochemical recurrence, with many opting for modifications of these instead.

Further variation in patient populations was noted in their prior treatments, with over half of the studies recruiting mixed populations of patients initially treated with radical prostatectomy and/or radiotherapy. Perhaps consistent with the observation of its lowest range of median PSA levels, a greater proportion of post-prostatectomy patients was noted for the ^18^F-flotufolastat studies than for the other radiopharmaceuticals.

Given these disparities in patient cohorts, we evaluated the patient-level detection rates in more homogenous populations by extracting any detection rate data stratified by patients’ prior treatment, or in patients with low (<1 ng/mL) PSA levels. When stratified by prior treatment, the sample-weighted mean data show high detection rates for each of the radiopharmaceuticals in post-radiotherapy patients, ranging from 87% for ^68^Ga-PSMA-11 to 99% for ^18^F-flotufolastat. As might be expected, lower rates were achieved in post-prostatectomy patients, ranging from 55% for ^18^F-DCFPyL to 76% for ^18^F-flotufolastat. Moreover, the sample-weighted mean data reveal that ^18^F-flotufolastat identified recurrence in two-thirds of patients with a PSA level < 1 ng/mL, compared with approximately half the patients scanned with ^68^Ga-PSMA-11 or just two-fifths with ^18^F-DCFPyL, with similarly low-level PSA recurrences.

The high detection rate of ^18^F-flotufolastat at low PSA levels has been noted previously [53], with the authors suggesting that its favorable biodistribution, with sustained plasma bioavailability and limited bladder activity at the point of imaging [10], offers a diagnostic advantage [53]. Furthermore, in contrast with other PSMA-targeted radiopharmaceuticals, ^18^F-flotufolastat shows high-affinity receptor binding and internalization which likely contribute to its high detection rate [54].

Aside from the heterogeneity noted across the patient populations of the included studies, the main limitation of these data is that the patient-level detection rate evaluated here provides the overall percent positivity of patients scanned with the PSMA-targeted radiopharmaceutical in question. These positive PET findings were not verified pathologically and so likely include a proportion of false positive findings along with the true positive results of imaging. As indicated by the QUADAS-2 assessment, the risk of bias in the reference standard domain is considered high. Typical of studies in patients with recurrent prostate cancer, there are multiple practical and ethical factors which limit histopathological verification of the PET findings, especially when the scan is negative. Moreover, histopathology is unlikely to accurately determine every site of metastatic disease. While some studies attempted to verify positive scans by a variety of methods including histopathology, conventional imaging, and PSA follow up monitoring in order to verify PSMA-positive lesions, only 11% of the included studies reported a PPV estimate. Moreover, as recently reported [55], the standard of truth methodology used to verify PSMA-PET findings substantially impacts estimates, which could impact cross-trial comparisons. However, our analysis did identify studies reporting patient-level PPV estimates for each of the three evaluated radiopharmaceuticals using the gold standard histopathological verification of PET-positive detections. These data show very similar PPV estimates for the three radiopharmaceuticals (84% for ^68^Ga-PSMA-11 [28], 79–83% for ^18^F-DCFPyL [47], and 82% for ^18^F-flotufolastat [11]), and so it might be assumed that for each radiopharmaceutical a similar proportion of PET-positive lesions will be verified as true positive on a patient-level.

Vastly more studies were available for ^68^Ga-PSMA-11 than for ^18^F-DCFPyL or ^18^F-flotufolastat, likely a reflection of ^68^Ga-PSMA-11 having the earliest FDA-approval date [5,6,7]. Only two studies were available for ^18^F-flotufolastat, with the majority of data coming from a phase 3 registration trial, which had more stringent inclusion parameters than some of the post-approval studies reported for the earlier approved agents. Our analysis included only studies with larger patient cohorts (≥ 100 patients) to avoid the variability in data from small sample sizes. However, the relative paucity of large scale ^18^F-DCFPyL and ^18^F-flotufolastat studies limit our findings and further systematic literature reviews are warranted at a later date.

## 5. Conclusions

Considerable heterogeneity exists across populations in studies of diagnostic PET radiopharmaceuticals. Although we note a relative paucity of ^18^F-DCFPyL and ^18^F-flotufolastat studies compared with ^68^Ga-PSMA-11, the available data suggest that all three radiopharmaceuticals have high detection rates when considering the alternative conventional imaging modalities that were most commonly used previously. Due to the inherent differences in inclusion criteria and heterogeneity amongst the studies, we cannot make any conclusions around comparative detection rates. Head-to-head trials would provide direct comparisons to determine potential superiority of any one PSMA-targeted radiopharmaceutical over another. Until such studies are performed, decisions to use a specific radiopharmaceutical will be likely be determined by access to these agents, ease of administration, and discretion of the ordering physicians.

## Figures and Tables

**Figure 1 diagnostics-15-01224-f001:**
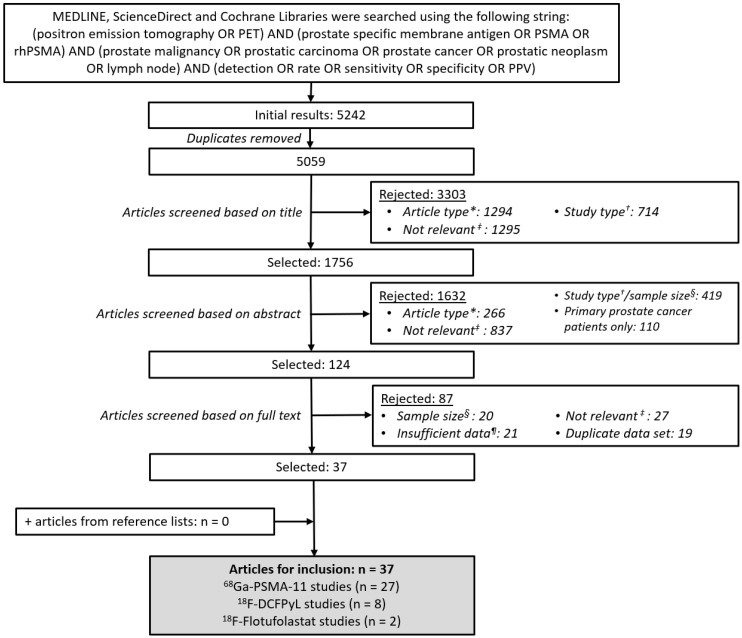
PRISMA flow chart illustrating the systematic literature search process. * Excluded articles comprised review articles, practice guidelines, editorials, and letters. ^†^ Excluded study types included case studies, case series and preclinical evaluations. ^‡^ Excluded studies included those evaluating a radiopharmaceutical other than ^68^Ga-PSMA-11, ^18^F-DCFPyL, or ^18^F-flotufolastat, those focused on another cancer type, therapeutic studies, and any studies that included patients with PSA persistence following primary therapy among their cohort of patients with biochemical recurrence. ^§^ Any studies that reported data from fewer than 100 patients in the cohort of interest were excluded. ^¶^ The minimum data requirement for inclusion was for a patient-level ^68^Ga-PSMA-11, ^18^F-DCFPyL, or ^18^F-flotufolastat detection rate in 100 or more patients with biochemical recurrence of prostate cancer.

**Figure 2 diagnostics-15-01224-f002:**
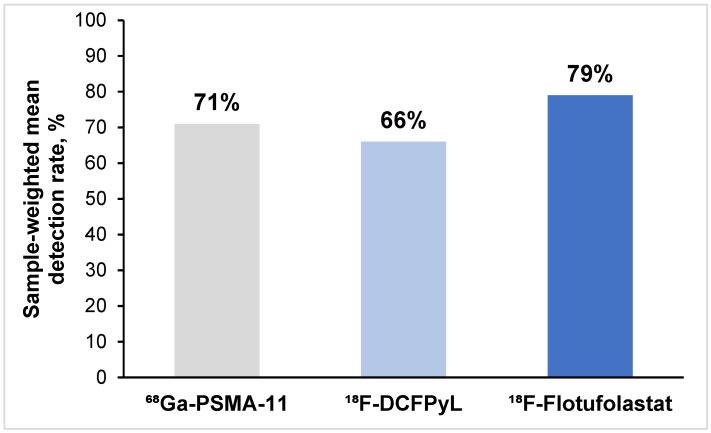
Sample-weighted mean patient-level detection rates across the 27 ^68^Ga-PSMA-11 studies, 8 ^18^F-DCFPyL studies, and 2 ^18^F-flotufolastat studies.

**Figure 3 diagnostics-15-01224-f003:**
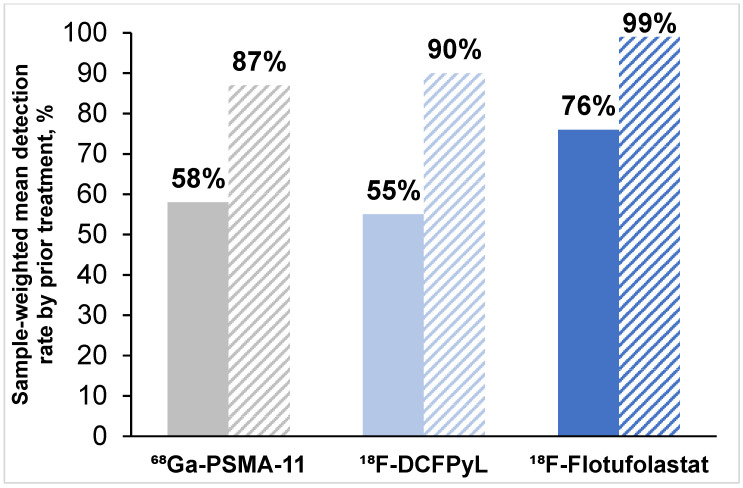
Sample-weighted mean patient-level detection rates across the included studies stratified by prior treatment. Post-prostatectomy data (solid bars) were available for 12 ^68^Ga-PSMA-11 studies, 4 ^18^F-DCFPyL studies and 2 ^18^F-flotufolastat studies. Post-radiotherapy data (hatched bars) were reported for 4 ^68^Ga-PSMA-11 studies, 2 ^18^F-DCFPyL studies, and 1 ^18^F-flotufolastat study.

**Figure 4 diagnostics-15-01224-f004:**
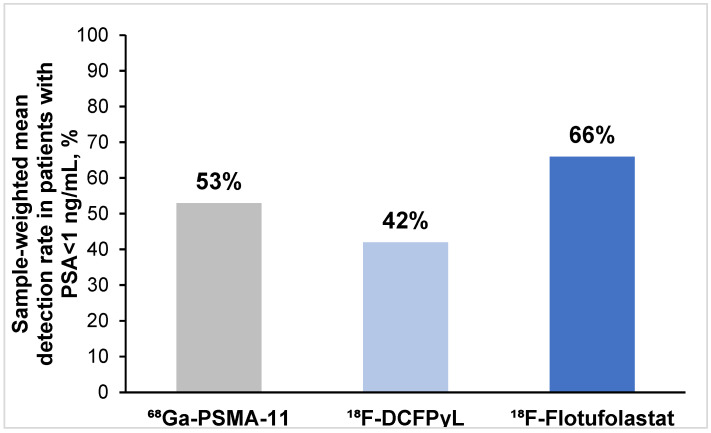
Sample-weighted mean detection rates in patients with PSA levels < 1 ng/mL across the included studies. Detection rates among patients with PSA < 1 ng/mL were reported for 13 ^68^Ga-PSMA-11 studies, 5 ^18^F-DCFPyL studies, and 2 ^18^F-flotufolastat studies.

**Figure 5 diagnostics-15-01224-f005:**
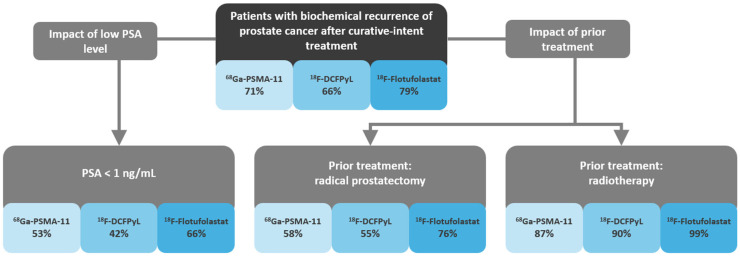
Summary of sample-weighted mean detection rates and impact of low PSA and prior treatment.

**Table 1 diagnostics-15-01224-t001:** Risk of bias appraisal of the included studies according to the QUADAS-2 tool.

Article	Risk of Bias			
	Patient Selection	Index Test	Reference Standard	Flow and Timing
Akdemir, 2019 [16]	LOW	LOW	HIGH	HIGH
Alberts, 2022 [17]	LOW	LOW	HIGH	HIGH
Armstrong, 2024 [18]	LOW	LOW	HIGH	HIGH
Boreta, 2019 [19]	LOW	LOW	HIGH	HIGH
Brito, 2019 [20]	LOW	LOW	HIGH	HIGH
Burgard, 2023 [21]	UNCLEAR	LOW	HIGH	HIGH
Caroli, 2018 [22]	LOW	LOW	HIGH	HIGH
Cerci, 2022 [23]	UNCLEAR	LOW	HIGH	HIGH
Christensen, 2022 [24]	LOW	LOW	HIGH	HIGH
Einspieler, 2017 [25]	LOW	LOW	HIGH	HIGH
Emmett, 2020 [26]	UNCLEAR	LOW	HIGH	HIGH
Farolfi, 2019 [27]	UNCLEAR	LOW	HIGH	HIGH
Fendler, 2019 [28]	LOW	LOW	HIGH	HIGH
Fourquet, 2021 [29]	LOW	LOW	HIGH	HIGH
Freitag, 2017 [30]	LOW	LOW	HIGH	HIGH
Hamed, 2019 [31]	LOW	LOW	HIGH	HIGH
Hoffmann, 2020 [32]	LOW	LOW	HIGH	HIGH
Lawal, 2021 [33]	UNCLEAR	LOW	HIGH	HIGH
McCarthy, 2019 [34]	UNCLEAR	LOW	HIGH	HIGH
Miksch, 2020 [35]	UNCLEAR	LOW	HIGH	HIGH
Müller, 2019 [36]	LOW	LOW	HIGH	HIGH
Pinot, 2023 [37]	LOW	LOW	HIGH	HIGH
Rauscher, 2018 [38]	UNCLEAR	LOW	HIGH	HIGH
Rauscher, 2020 [39]	LOW	LOW	HIGH	HIGH
Raveenthiran, 2019 [40]	LOW	LOW	HIGH	HIGH
Seniaray, 2020 [41]	LOW	LOW	HIGH	HIGH
Verburg, 2016 [42]	LOW	LOW	HIGH	HIGH
García-Zoghby, 2023 [43]	UNCLEAR	LOW	HIGH	HIGH
Li, 2025 [44]	UNCLEAR	LOW	HIGH	HIGH
Meijer, 2020 [45]	LOW	LOW	HIGH	HIGH
Mena, 2022 [46]	UNCLEAR	LOW	HIGH	HIGH
Morris, 2021 [47]	UNCLEAR	LOW	HIGH	HIGH
Oprea-Lager, 2023 [48]	LOW	LOW	HIGH	HIGH
Rousseau, 2019 [49]	LOW	LOW	HIGH	HIGH
Sigurdson, 2024 [50]	LOW	LOW	HIGH	HIGH
Jani, 2023 [11]	LOW	LOW	HIGH	HIGH
Rauscher, 2021 [51]	LOW	LOW	HIGH	HIGH

**Table 2 diagnostics-15-01224-t002:** Patient characteristics and overall detection rates across the included studies.

Article	Radiopharmaceutical Evaluated	Number of Patients in the Relevant Cohort	Median PSA, ng/mL	Overall Patient-Level Detection Rate ^a^, %	Proportion of Patients Who Had Undergone Radical Prostatectomy, %
Akdemir, 2019 [16]	^68^Ga-PSMA-11	121	3.9	76	47
Alberts, 2022 [17]	^68^Ga-PSMA-11	122	2.8	87	79
Armstrong, 2024 [18]	^68^Ga-PSMA-11	102	0.2	38	100
Boreta, 2019 [19]	^68^Ga-PSMA-11	125	0.4	53	100
Brito, 2019 [20]	^68^Ga-PSMA-11	100	1.7	72	69
Burgard, 2023 [21]	^68^Ga-PSMA-11	115	0.1	25	100
Caroli, 2018 [22]	^68^Ga-PSMA-11	314	0.8	63	84
Cerci, 2022 [23]	^68^Ga-PSMA-11	1004	NR (mean = 1.6)	65	78
Christensen, 2022 [24]	^68^Ga-PSMA-11	189	10.5	55	81
Einspieler, 2017 [25]	^68^Ga-PSMA-11	118	6.4	91	0
Emmett, 2020 [26]	^68^Ga-PSMA-11	260	0.3	65	100
Farolfi, 2019 [27]	^68^Ga-PSMA-11	119	0.3	34	100
Fendler, 2019 [28]	^68^Ga-PSMA-11	635	2.1	75	73
Fourquet, 2021 [29]	^68^Ga-PSMA-11	294	NR (mean = 3.0)	69	86
Freitag, 2017 [30]	^68^Ga-PSMA-11	119	1.7	78	100
Hamed, 2019 [31]	^68^Ga-PSMA-11	188	2.2	88	38
Hoffmann, 2020 [32]	^68^Ga-PSMA-11	660	10.7	76	81
Lawal, 2021 [33]	^68^Ga-PSMA-11	247	2.7	81	64
McCarthy, 2019 [34]	^68^Ga-PSMA-11	238	2.6	77	64
Miksch, 2020 [35]	^68^Ga-PSMA-11	116	NR (mean = 0.3)	50	100
Müller, 2019 [36]	^68^Ga-PSMA-11	223	1.0	74	98
Pinot, 2023 [37]	^68^Ga-PSMA-11	159	0.8	66	90
Rauscher, 2018 [38]	^68^Ga-PSMA-11	272	0.5	65	100
Rauscher, 2020 [39]	^68^Ga-PSMA-11	102	0.9	80	100
Raveenthiran, 2019 [40]	^68^Ga-PSMA-11	276	3.6	86	0
Seniaray, 2020 [41]	^68^Ga-PSMA-11	124	1.8	70	100
Verburg, 2016 [42]	^68^Ga-PSMA-11	155	4.0	80	64
García-Zoghby, 2023 [43]	^18^F-DCFPyL	138	NR (mean = 2.8)	65	57
Li, 2025 [44]	^18^F-DCFPyL	415	0.4	49	100
Meijer, 2020 [45]	^18^F-DCFPyL	262	2.5	86	48
Mena, 2022 [46]	^18^F-DCFPyL	245	1.6	79	80
Morris, 2021 [47]	^18^F-DCFPyL	208	0.8	62	85
Oprea-Lager, 2023 [48]	^18^F-DCFPyL	201	0.5	58	73
Rousseau, 2019 [49]	^18^F-DCFPyL	130	NR (mean = 5.2)	85	72
Sigurdson, 2024 [50]	^18^F-DCFPyL	116	0.2	52	100
Jani, 2023 [11]	^18^F-Flotufolastat	389	1.1	83	79
Rauscher, 2021 [51]	^18^F-Flotufolastat	242	0.6	73	100

^a^ Either reported directly or extrapolated from data included in the article. NR, not reported.

## Data Availability

The data reported in this article are available from the corresponding author on reasonable request.

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
