# Peer review of "Detection Rates of PSMA-PET Radiopharmaceuticals in Recurrent Prostate Cancer: A Systematic Review"

_diagnostics, 2025, doi:10.3390/diagnostics15101224_

Round 1
Reviewer 1 Report
Comments and Suggestions for Authors
this is a well planned and written sistematic review on PSMA-targeting compounds in Pca patients. I Particularly appreciated the head-to-head comparison of detection rates among different radiopharmaceuticals, the decision to choose studies including more than 100 participants and to stratify by PSA levels, pointing that after PSMA, PSA levels for defyning BCR have to be reviewed.
A better description of biodistribution of Ga-PSMA versus fluorinated ones could improve the understanding of the better detection rate of GaPSMA in low PSA rerum levels.
This review lacks only false positive reporting with different RP, but this was not aimed by the authors.
Reviewer 2 Report
Comments and Suggestions for Authors
Reviewer Comments to the Authors
The Manuscript makes an important contribution to the literature on PSMA PET imaging. The authors have done a thorough job synthesizing detection rates across different radiopharmaceuticals (a topic that's become increasingly popular in clinical practice). The PRISMA-guided methodology appears sound, and the analyses stratified by PSA levels and prior treatments. These nuances will help when we are deciding which tracer to use for specific patient scenarios.
Major Comments:
Heterogeneity Concerns:
While the authors appropriately note the heterogeneity between studies (something I've struggled with when interpreting this literature myself), I would encourage them to go further in discussing how these differences affect our ability to compare tracers. The varying PSA thresholds and treatment histories are particularly problematic when reading. Would a sensitivity analysis help? For example, excluding studies with PSA >10 ng/mL might clarify whether 68Ga-PSMA-11's performance holds in lower PSA ranges.
The QUADAS-2 results are somewhat worrisome - the high risk of bias in reference standards is a fact that without histopathological confirmation, it can not be certain whether higher detection rates reflect true sensitivity or false positives. This issue deserves more emphasis in the discussion.
Questions About 18F-flotufolastat Data:
There is only two studies for 18F-flotufolastat. Having reviewed the cited papers, I believe these are both controlled trials (correct me if I'm wrong). The real-world performance often differs from trial results - did the authors consider how stricter inclusion criteria might make 18F-flotufolastat appear better than it performs in routine practice?
Suggestions for Improvement:
- A visual comparison of tracer performance (perhaps with confidence intervals) would help readers quickly grasp the key differences
- The superior low-PSA performance of 18F-flotufolastat is intriguing ( might this relate to its reportedly lower urinary activity?). Some speculation on mechanisms could enrich the discussion.
It seems considering adding a table summarizing the practical pros and cons of each tracer based on the evidence would help to grasp nuances.
This is important work that deserves publication after addressing these points. As someone who regularly uses some of these tracers, I found the analysis thoughtful and clinically relevant, though the limitations (particularly the lack of head-to-head trials) better be clearly emphasized to guide appropriate interpretation.
